# Chamber with Inverted Electrode Geometry for Measuring and Control of Ion Flux-Energy Distribution Functions



**Christian Schulze** [1] , **He Li** [1] , **Leonie Mohn** [1] , **Martin Müller** [2] and **Jan Benedikt** [1,*]

1   Institute of Experimental and Applied Physics, Kiel University, Leibnitzstr 19, D-24098 Kiel, Germany; schulze@physik.uni-kiel.de (C.S.); heli@physik.uni-kiel.de (H.L.); leonie.mohn@googlemail.com (L.M.)
2   Institute of Physics, Czech Academy of Sciences, Cukrovarnická 10, 182 21 Prague, Czech Republic; mullerm@fzu.cz
*   Correspondence: benedikt@physik.uni-kiel.de; Tel.: +49-(0)431-880-3879

**Abstract:** Measurements of ion flux-energy distribution functions at the high sheath potential of the driven electrode in a classical low-pressure asymmetric capacitively coupled plasma are technically difficult as the diagnostic device needs to float with the applied radio frequency voltage. Otherwise, the ion sampling is disturbed by the varying electric field between the grounded device and the driven electrode. To circumvent such distortions, a low-pressure plasma chamber with inverted electrode geometry, where the larger electrode is driven and the smaller electrode is grounded, has been constructed and characterized. Measurements of the ion flux-energy distribution functions with an energy-selective mass spectrometer at the high sheath potential of the grounded electrode are presented for a variety of conditions and ions. The potential for suppressing low-energy ions from resonant charge transfer collisions in the sheath by the dilution of the working gas is demonstrated. Additionally, the setup is supplemented by an inductively coupled plasma that controls the plasma density and consequently the ion flux to the substrate while the radio frequency bias controls the ion energy. At high ion energies, metal ions are detected as a consequence of the ionization of sputtered electrode material. The proposed setup opens a way to study precisely the effects of ion treatment for a variety of substrates such as catalysts, polymers, or thin films.

**Keywords:** capacitively coupled plasma; inductively coupled plasma; rf bias; ion energy distribution; energy-selective mass spectrometry

## 1. Introduction

The composition and energy of ions reaching plasma-facing surfaces are key parameters in low-pressure plasma surface treatments because ion bombardment is the enabling feature of for example anisotropic and selective etching in the microelectronic industry, physical sputtering in deposition applications such as magnetron sputtering or high-power impulse magnetron sputtering (HiPIMS), and material doping in plasma immersion ion implantation (PIII) processes.

Energy selective mass spectrometry (ESMS) has become the standard diagnostic for the analysis of plasma-generated ions in low-pressure plasmas. It provides the flux-energy distribution functions (IEDFs) for selective ion masses measured at the substrate surface with an integrated sampling orifice. In contrast to residual gas analysis (RGA) mass spectrometry, the signal intensity is usually high because ions can be effectively collected by ion optics and no ionization process inside the mass spectrometer is needed. Every peak in the ion mass spectrum represents at least one ion with a particular mass-to-charge ratio $m/q$ originating from the plasma, so no fragmentation occurs after the ion sampling as is the case in RGA measurements of neutral species. There is also no background signal, making the signal analysis straightforward. The ESMS is, however, providing only relative signals, since the absolute flux calibration is difficult due to an unknown acceptance angle and a

mass and energy-dependent ion transmission function through the mass spectrometer [1]. Absolute energy selective ion currents to a given substrate can be measured with the help of the retarding field analyzer (RFA), which, however, does not provide mass selectivity. ESMS has been used for the characterization of various plasma sources including dc and rf plasmas, HiPIMS discharges and ion beam experiments [2].

In most technical applications, rf (biased) plasmas are widely used since either insulating/semiconducting substrates are treated or the applied coating materials are non-conducting, which leads in both cases to charging effects in dc-driven plasma sources. Furthermore, high ion energies are required for most of the applications. However, measurements of the IEDFs at the high sheath potential of the rf-powered electrode are challenging, since the ions are influenced by the rf field between the sampling orifice at rf potential and the mass spectrometer's ion lenses that are typically defined with respect to ground resulting in possible artifacts in the IEDF [3–5]. This effect can be avoided with the technically difficult solution of a mass spectrometer floating with the rf potential as it has been performed by Mizutami et al. [4–6]. In our opinion, the most elegant solution has been provided by Wild and Koidl [7,8] with an inverted electrode geometry, where the surface ratio of the voltage driven and grounded electrode is reversed, so the plasma facing surface of the driven electrode is much larger than the surface of the grounded electrode. The capacitive coupling requires an equal electron and ion current averaged over the rf cycle which is achieved in this case by the formation of a positive dc offset at the rf-driven electrode. Since the potential of the larger electrode corresponds to the plasma potential, the larger sheath potential is observed across the sheath in front of the grounded electrode, where IEDFs can be measured without perturbations. Wild and Koidl were able to measure IEDFs with the help of the RFA integrated into the grounded electrode.

Inspired by this work, we have constructed a similar chamber with inverted electrode geometry with the sampling orifice of an energy-selective mass spectrometer integrated into the small grounded electrode and with the rf bias applied to the chamber wall. Our measurement clearly demonstrates that this configuration enables undisturbed and straightforward measurements of IEDFs. Additionally, we have studied the possibility of modifying the IEDF of argon ions by changing the gas mixture. Finally, the concept of the reactor with inverted electrode geometry has been extended by the implementation of inductive plasma coupling to achieve independent control of plasma density and ion energy.

## 2. Materials and Methods

We used a setup that applies a capacitively coupled 13.56 MHz sinusoidal rf voltage directly to the vacuum chamber wall as sketched in Figure 1 for the measurements presented here. The smaller electrode with an integrated sampling orifice is electrically isolated from the vacuum chamber and mounted to a grounded energy-selective mass spectrometer. The surface ratio of the driven and grounded electrode is about 20 to ensure a highly asymmetric electrode geometry. Special care must be taken to isolate all external connections that might ground the vacuum chamber such as the gas supply, pump connection, and pressure measurement unit. This is realized via sets of two grids at each connecting flange, where one grid is at the grounded part and the other at the rf-powered chamber wall. These grids are at a distance of 1 mm, below the typical sheath width of the plasma in the chamber, avoiding hence the formation of parasitic discharges at these connecting ports. Furthermore, the vacuum chamber is enclosed inside a grounded shielding cage (not shown in the Figure 1) to prevent the radiation of the rf power. The voltage applied to the chamber wall is measured with a high-voltage probe (Tektroniks P5100A).

In addition to the capacitively coupled plasma (CCP), a quartz cylinder and a planar coil have also been mounted on top of the chamber to ignite an inductively coupled plasma (ICP) as sketched in Figure 2. Then, the CCP rf power works as an rf bias that controls the sheath potential and consequently the ion energy but not the plasma density. In this setup, the ICP coil is driven separately with an additional matching circuit and rf power

supply, that is synchronized to the CCP power supply to reduce instabilities. The ICP coil is powered with 100 W in the experiments discussed here.

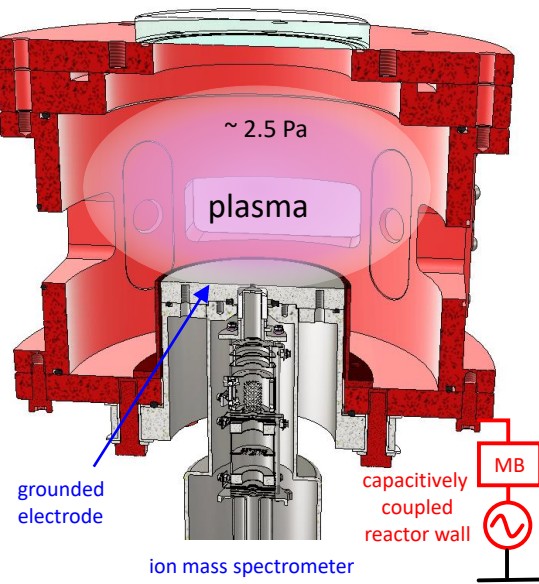

**Figure 1.** Sketch of the chamber with inverted electrode geometry. The red color represents the parts driven by the rf voltage, grounded parts are sketched in grey. The parts are electrically isolated by a 1 mm PTFE spacer. Some windows, screws, other small parts, and the grounded shielding cage are not shown.

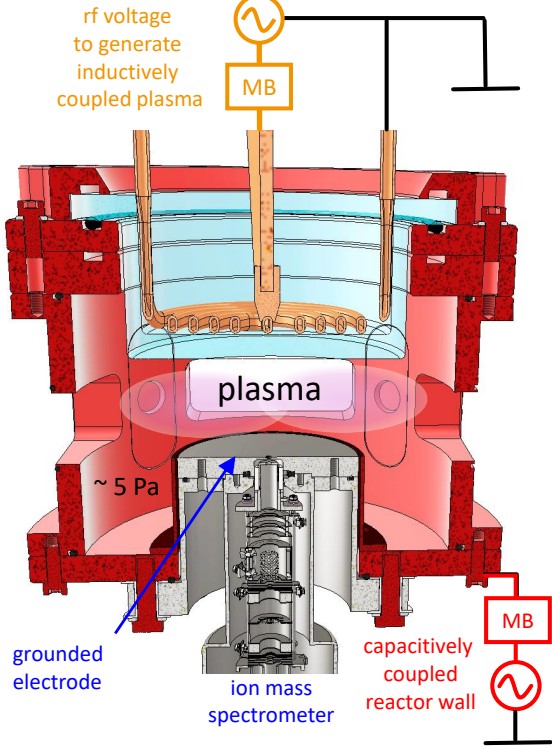

**Figure 2.** Sketch of the chamber with inverted electrode geometry and with an additional planar coil (orange) in quartz dome (blue) to maintain inductive power coupling. The red color represents the parts driven by the rf voltage, grounded parts are sketched in grey. The parts are electrically isolated by a 1 mm PTFE spacer. Some windows, screws, other small parts, and the grounded shielding cage are not shown.

The chamber is evacuated by a two-stage pumping system (turbomolecular pump and membrane pump) to a base pressure of about $5 \times 10^{-4}$ Pa. Mass flow controllers deliver a constant gas flow of 1 sccm. The pressure is controlled via a butterfly valve in the pressure range of about 1 Pa to 40 Pa.

A commercial energy selective mass spectrometer (Hiden Analytical PSM003) is used to measure the ion flux that passes the 100 µm orifice in the grounded electrode. It offers energy selection by means of a Bessel-box energy filter in the energy range of up to 1000 eV with an energy resolution of 0.7 eV. The subsequent quadrupole mass filter selects ions of up to 300 amu. A secondary electron multiplier with a dynamic range of 7 decades is used for ion detection. The setup is differentially pumped to about $1 \times 10^{-5}$ Pa during operation.

### 3. Results and Discussion

#### 3.1. Ion Analysis in an Ar CCP

The formation of different IEDFs for different ion species is exemplary shown for an Ar CCP at 300 $V_{pp}$ and 2.5 Pa chamber pressure. To analyze the ion composition in the plasma, mass spectra have been measured at several ion energies to identify the dominant ions hitting the grounded electrode. Argon-related ions such as $Ar^{2+}$, $Ar_2^+$, and $ArH^+$ were detected in addition to the dominant $Ar^+$ ion as well as low-density ions that are related to impurities such as $H_2^+$, $H_2O^+$ or $N_2^+$. A typical mass spectrum is discussed in Section 3.3 in more detail.

Figure 3 presents the IEDFs of the argon-related ions $Ar^{2+}$ (20 amu), $Ar^+$ (40 amu), $ArH^+$ (41 amu), and $Ar_2^+$ (80 amu). Please note that ESMS distinguishes the ion's mass-to-charge ratio ($m_i/q_i$ in amu/$e$) and energy-to-charge ratio ($E_i/q_i$ in eV/$e$). For example, doubly ionized argon ($Ar^{2+}$) appears at 20 amu/$e$ in the mass spectrum and has twice the energy shown on the energy scale (Imagine it as two halves of the argon atoms moving together, each with elementary charge $e^+$, mass 20 amu and the energy shown on the energy scale). Since all ions except $Ar^{2+}$ are singly ionized, the term ion mass ($m_i$ in amu) and ion energy ($E_i$ in eV) is used throughout this work for convenience.

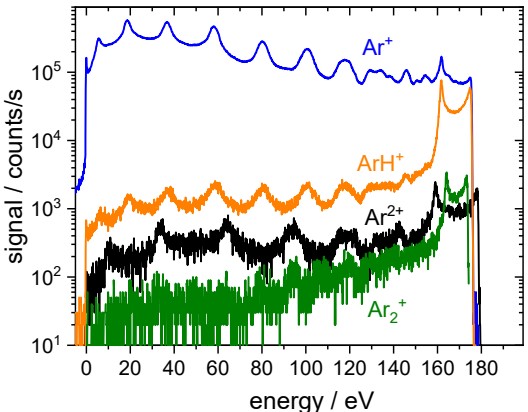

**Figure 3.** Measured IEDFs for argon-related ions in an Argon discharge at 2.5 Pa and 300 $V_{pp}$.

In low-pressure rf plasmas, the IEDF at the electrodes is a superposition of a bimodal peak structure around the time-averaged sheath potential [3,9–11] generated by ions traversing the sheath without collisions and a multiple peak structure at lower energies [6–8,12–15] as a result of ion collisions in the sheath. The dominance of one of these structures is determined by the amount of decelerating ion-neutral collisions and, therefore, related to the ion collision cross section, the density of the collisional partners in the sheath, and the sheath width. In addition, ion formation due to electron impact ionization and dissociative recombination of molecular ions in the sheath might play a role but it is expected to be negligible due to the low electron density in the sheath. For most ions, the dominant ion neutral interaction is the resonant charge transfer collision or charge exchange collision (CXC) in which a fast ion collides with a slow neutral particle and transfers its charge to it

resulting in fast neutrals and slow ions. CXC of ions with the neutral species of the same art is resonant with a large value of the CXC cross section. For an argon plasma:

$$\mathrm{Ar}_{\mathrm{fast}}^+ + \mathrm{Ar}_{\mathrm{slow}} \rightarrow \mathrm{Ar}_{\mathrm{fast}} + \mathrm{Ar}_{\mathrm{slow}}^+$$

Depending on the position in the sheath and phase of the rf voltage, at which these slow ions are generated, they are accelerated only in a fraction of the sheath potential and gain, therefore, less energy as they reach the substrate.

The bimodal peak structure result from ions that enter the sheath at different phases of the rf cycle. The separation of the two bimodal peaks is related to the ratio of the rf cycle duration $\tau_{\mathrm{rf}}$ and ion transit time through the sheath $\tau_{\mathrm{i}}$

$$\Delta E = \frac{4\,\tau_{\mathrm{rf}}}{\pi\,\tau_{\mathrm{i}}} e\,\Delta\Phi_{\mathrm{s}} \tag{1}$$

with the variation of the sheath potential during an rf cycle $\Delta\Phi_{\mathrm{s}}$ [16–20]. In our case of highly asymmetric electrode geometry, the sheath potential varies from a low floating potential (about 20 eV) to the rf peak-to-peak voltage plus the floating potential during an rf cycle, so $\Delta\Phi_{\mathrm{s}}$ equals the peak-to-peak voltage. The ion transit time itself depends on sheath width $s$, ion mass $m_{\mathrm{i}}$ and the time-averaged sheath potential $\overline{\Phi}_{\mathrm{s}}$

$$\tau_{\mathrm{i}} = 3s\sqrt{\frac{m_{\mathrm{i}}}{2e\overline{\Phi}_{\mathrm{s}}}} \tag{2}$$

According to the above-mentioned variation of the sheath potential, the average sheath potential is about 20 eV higher than the rf amplitude, so in the case of Figure 3 with 300 $V_{\mathrm{PP}}$, at about 170 eV.

The IEDFs of $\mathrm{Ar}^{2+}$, $\mathrm{ArH}^+$ and $\mathrm{Ar}_2^+$ are dominated by ions traversing the sheath without collisions, and the bimodal peaks centered around the energy that corresponds to the ion acceleration in the time averaged-sheath potential are observed. This is the consequence of the low density of the neutral collision partners that are necessary for resonant CXC.

On the contrary, the IEDF of $\mathrm{Ar}^+$ in Figure 3 shows mainly peaks at lower energies from collisions in the sheath, whereas its bimodal peak structure is only a minor feature. These low-energy peaks originate in resonant CXC in the sheath [7,8,13–15] as the cross section for $\mathrm{Ar}^+$-Ar resonant CXC is large [21], the density of the collision partner (neutral Ar) is high, and the sheath width is also large due to the relatively low plasma density (large Debye length).

To demonstrate the influence of density on the number of collisions, Figure 4 shows the IEDFs of $\mathrm{Ar}^+$ ions for pressures between 1.25 Pa and 40 Pa. As the amount of collisions increases with pressure, the bimodal peak structure diminishes. Only the IEDFs for pressures below 5.0 Pa show a significant amount of collision-free ions from the bulk plasma.

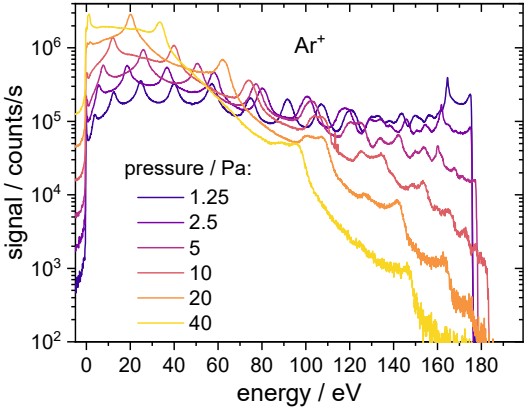

**Figure 4.** Dependence of the measured $\mathrm{Ar}^+$ IEDFs on the chamber pressure. Applied voltage 300 $V_{\mathrm{PP}}$.

### 3.2. Effect of Ar Gas Dilution with He

The large collision rate of Ar$^+$ ions in the sheath is determined both by the high argon atom density and the large CXC cross section. To reduce the effect of these resonant CXC collisions one has to lower the density (pressure) of the neutral argon atoms as demonstrated in the Figure 4. However, the number of collisions can be effectively reduced by diluting Ar with He as well. Helium has very large ionization and excitation thresholds and the argon gas is, therefore, preferentially ionized in the plasma. Additionally, CXCs are not resonant for collisions of Ar$^+$ with He, and the formation of slow He$^+$ ions in their mutual collisions is prevented by the large ionization energy of helium atoms. Additionally, the momentum transfer in elastic collisions of Ar$^+$ with He atoms will be also small due to the large mass difference between both species.

Figure 5 shows the IEDFs of Ar$^+$ and He$^+$ ions as measured at a pressure of 2.5 Pa for several Ar/He gas mixtures. Additionally, the Ar$^+$ IEDF of an argon discharge from Figure 3 is shown for comparison as well. In Figure 5a, the 90% dilution of argon gas with helium (argon partial pressure of 0.25 Pa) clearly suppresses the collisional peaks in the IEDFs in favor of the bimodal peak structure as seen in the direct comparison with the IEDFs measured in pure argon. The He$^+$ ions appear as well with collisional IEDFs but with relatively low signal intensities compared to argon. It should be further noted that the ion transmission probability function of a typical quadrupole mass spectrometer scales with the ion mass to the power of $x$, where $x$ is usually between $-0.5$ and $-1$, discriminating hence heavy ions against light ones [22–24]. Taking this mass discrimination into account for He$^+$ and Ar$^+$ will result in the relative decrease of the signal intensities measured for He$^+$ with a factor between 3 and 10. Absolute energy resolved ion flux measurements with RFA could provide the correct value of the discrimination factor by comparing ESMS and RFA measurements performed in pure Ar and pure He. A further dilution to 96.4% He (argon partial pressure of 0.09 Pa) suppresses the collisional peaks in the Ar$^+$ IEDFs even further as shown in Figure 5b. The intensity of the He$^+$ signal increases, but considering the previously discussed mass discrimination, the Ar$^+$ bimodal peak structure at around 180 eV will probably still dominate the ion flux to the grounded electrode.

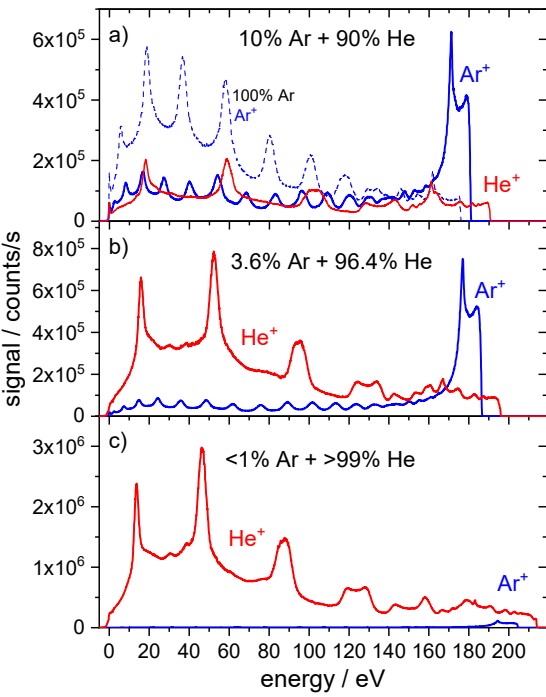

**Figure 5.** IEDFs of Ar$^+$ and He$^+$ measured at several Ar/He gas mixtures at gas pressure of 2.5 Pa and applied voltage of 300 V$_{pp}$.

For Figure 5c, the Ar gas flow is switched off and only the leakage of the mass flow controller supplies an unknown but very low residual flow of Ar. The remaining Ar concentration is too low to still dominate the ion composition and the discharge is finally controlled by helium. Hence the ion flux to the surface is ruled by the collisional He$^+$ IEDF and the Ar$^+$ IEDF shows only a weak signal from the bimodal peak structure. Nevertheless, we showed that the dilution of the Ar working gas with He in the order of 90% to 95% He can significantly reduce the fraction of low energy Ar$^+$ ions and can realize conditions with a well-defined, almost monoenergetic Ar$^+$ IEDF also for the low ion fluxes of a CCP discharge. Such an IEDF is preferred since better control over the surface processes can be achieved.

The positive effect of Ar dilution with He is due to the large difference in the ionization energy between Ar (15.76 eV) and He (24.59 eV). Using a very small admixture of Kr or Xe gases with ionization energies of just 14.00 eV or 12.13 eV [25] respectively, in place of Ar, would allow for even higher He dilutions with less flux of low energy He$^+$ ions due to the lower electron temperature needed for an effective ionization of Kr/Xe. Furthermore, according to (1) and (2), the energy spread of the bimodal peak structure $\Delta E$ scales with $m_i^{-0.5}$, so having ions with higher mass narrows this structure, making the IEDF more monoenergetic. Since only a very small fraction of the gas mixture consist of the expensive Kr or Xe gas, using He/Kr or He/Xe mixtures is a viable way in large-scale industrial applications, where precise ion treatment with a specific ion mass and narrow IEDFs are required.

### 3.3. Implementation of the Inductive Plasma Coupling

The combination of inductive and capacitive coupling of the rf power into the plasma is one possible solution enabling independent control of plasma density/ion flux and sheath voltage, where the high-density plasma is sustained by the inductive coupling with a rather low sheath potential, and the ion energy bombarding the surface is independently controlled by an additional dc self-bias formed by the capacitive coupling. This principle is used here, where the capacitive coupling is applied to the large area reactor wall resulting in a negative dc offset voltage on the small area grounded electrode. The ESMS integration of the grounded electrode as shown in the experimental setup in Figure 2 is used to analyze the changes in the IEDFs.

When applying only the rf voltage to the rector wall, similar ions with similar IEDFs as in Figure 3 are observed for an argon CCP at 5 Pa in Figure 6a. The applied power of 20 W corresponds to lower plasma density and lower sheath voltage than previously used, resulting in both lower ion energies and lower measured signals. Applying only inductive coupling with the power of 100 W as shown in Figure 6b results in large ion densities (large ion fluxes) with a single peak at an energy of about 12 eV corresponding to the ICP's plasma potential. Please note that almost no Ar$^{2+}$ ions are generated in a pure ICP as the maximum electron energies are low. In a CCP, secondary electrons can be accelerated in a high sheath potential to energies that easily exceed the second ionization energy of Ar (27.63 eV) [25]. This is not the case in ICPs where the sheath potential is much lower (12 eV in our case). Applying both inductive and capacitive coupling results in high plasma densities and ion fluxes at high ion energies due to the additional rf bias as shown in Figure 6c. In contrast to the low sheath potential in Figure 6b, the high sheath potential can effectively accelerate secondary electrons to high energies which results in a comparably high density of Ar$^{2+}$ ions. The higher plasma density also leads to a lower sheath width, which reduces the number of collisions in the sheath (the multiple peak structure at lower energies disappeared and the IEDF is dominated by the bimodal peak structure) and shorten significantly the ion transit time $\tau_i$ (c.f. (2)). The latter increases the energy spread of the bimodal peak structure $\Delta E$ (c.f. (1)) significantly. Ion treatments are less selective and less precise with such a broad bimodal peak structure. To narrow the bimodal peak structure under the conditions of high ion densities and low sheath width, the rf-bias frequency can be increased.

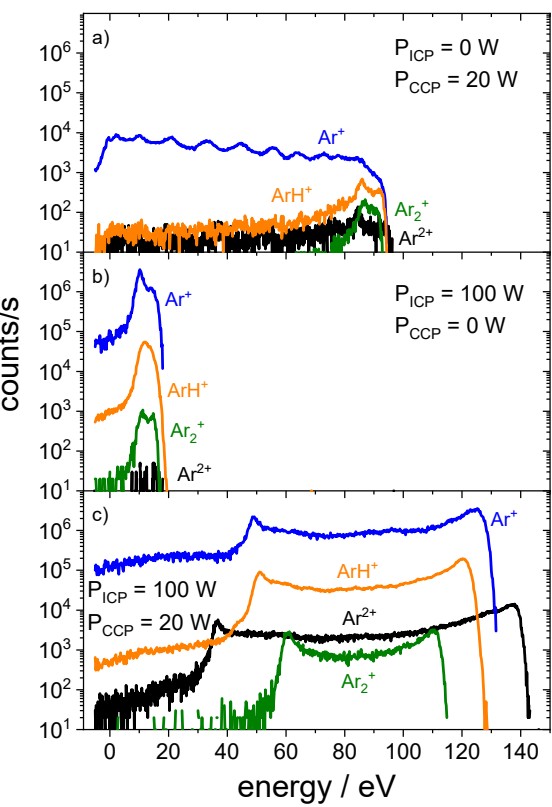

**Figure 6.** IEDFs of argon-related ions in an argon plasma at 5 Pa. (**a**) Only capacitive power coupling to the reactor wall with a power of 20 W. (**b**) Only inductive coupling with a power of 100 W applied to the planar rf coil. (**c**) Both inductive and capacitive couplings are applied.

The CCP power controls the sheath voltage and is directly linked to the ion energy as demonstrated in Figure 7. The measurements with the highest applied rf bias allow the detection and analysis of IEDFs not only for the argon ions but also for the ions of the sputtered material itself. Figure 8 shows ion mass spectra for the ICP and the rf-biased ICP measured at the energy in the center of the bimodal peak structure (12 eV for the ICP, 82 eV for the rf-biased ICP). While the ion composition in the ICP consists, besides the Ar-related ions, only of ions from impurities (containing H, C, N, and O), the mass spectrum of the rf-biased ICP shows additional peaks due to ionized atoms sputtered from the stainless steel electrode. The application of the rf bias results in physical sputtering of the stainless steel electrode and the formation of metal ions such as $Fe^+$, $Cr^+$, $Ni^+$, and $Mn^+$. The accumulation of metal ions in the discharge is also observed in the change of the plasma emission close to the grounded electrode that becomes blue when the rf bias is applied as shown in the insets in Figure 8. Even though sputtered metal ions are detectable in the ion flux, the sputtering damage to the electrode is negligible as the sputter rate is low for the typically used ion energies in the range of up to about 140 eV in this work. The sputtered material also coats the quartz cylinder, which shows a reflective but still semitransparent metal layer after several hours of operation. To ensure an unhindered rf field transmission, the cylinder is cleaned with nitric acid from time to time.

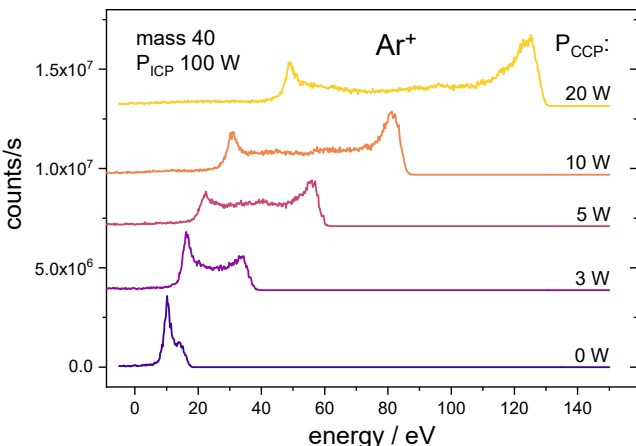

**Figure 7.** IEDFs of $Ar^+$ as function of applied power of the capacitively coupled bias on the reactor wall. Argon pressure 5 Pa.

The combination of the rf-biased ICP with the inverted electrode geometry and the ESMS diagnostics allows us to study in great detail the effects of different ion species, ion fluxes and ion energies on a variety of substrates under realistic plasma treatment conditions, where, for example, sputtering of the substrate material can strongly influence the ion composition and plasma properties in general.

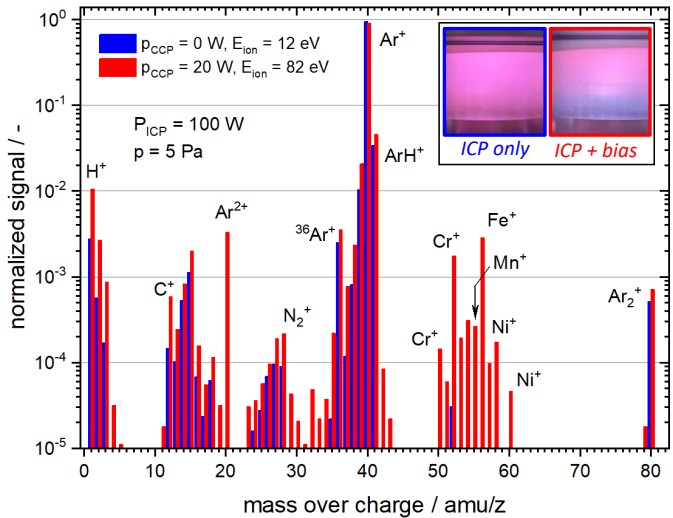

**Figure 8.** Mass spectrum measured in the ICP without bias (blue bars) at an ion energy of 12 eV compared to the mass spectrum measured under the ICP conditions with bias (red bars) measured at an ion energy of 82 eV (mean energy of the bimodal peak structure). Inset: photographs of the plasma between the grounded electrode and quartz glass dome for the two measured conditions.

## 4. Conclusions

We presented a concept for an rf CCP with inverted electrode geometry and an ESMS integrated into the grounded electrode. The shown measurements prove that the proposed setup allows for undisturbed IEDF measurements at the electrode with the larger sheath potential, which is in this case the grounded electrode. The measurements of the pressure dependence have shown an expected behavior of ion flux-energy distribution functions. $Ar^{2+}$, $ArH^+$, and $Ar_2^+$ have also been measured next to the dominant $Ar^+$ ion with similar results as known from the literature.

The effect of He admixture into the Ar on the IEDF of $Ar^+$ ions has been studied, where the argon dilution reduces the number of charge exchange collisions in the sheath.

We have shown that the plasma in gas mixtures with helium content between 90% and ≈96% are still dominated by argon ions, mainly due to the very large helium ionization energy, whereas the lower argon density at the constant overall pressure leads to a much longer $Ar^+$ mean free path in the sheath and a significant suppression of collisional low energy peaks in the $Ar^+$ IEDF. Argon dilution with helium is, therefore, a simple method of suppressing the low-energy ions in the IEDF, allowing for more defined and more selective ion-driven processes at the treated substrate.

Finally, the possibility to combine ICP and CCP in the reactor with inverted geometry has been successfully tested. We demonstrated the potential of this setup for future studies of ion interaction with a variety of materials, where the ICP power controls the plasma density and the capacitive coupling of the power to the reactor wall accelerates ions through the sheath. The ion mass spectrometry is used for the in situ monitoring of the ion energies and ion composition during the treatment, which is especially important for the treatments, where plasma surface interaction processes like sputtering or desorption strongly influence plasma properties.

**Author Contributions:** Conceptualization, J.B.; methodology, C.S. and J.B.; formal analysis, C.S. and J.B.; investigation, C.S., H.L., L.M. and M.M.; data curation, C.S., H.L., L.M. and M.M.; writing—original draft preparation, C.S. and J.B.; writing—review and editing, H.L. and M.M.; visualization, J.B.; supervision, M.M. and J.B.; project administration, J.B.; funding acquisition, J.B. All authors have read and agreed to the published version of the manuscript.

**Funding:** This work was supported by the German Research Foundation (DFG) via grant 426208229. H.L. thanks the China Scholarship Council (CSC) for the support through the project 202009110110. M.M. thanks for the support from CzechNanoLab Research Infrastructure supported by MEYS CR (LM2018110).

**Institutional Review Board Statement:** Not applicable.

**Informed Consent Statement:** Not applicable.

**Data Availability Statement:** The data that support the findings of this study are available upon reasonable request from the authors.

**Conflicts of Interest:** The authors declare no conflict of interest.

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
