# Peer review of "Chamber with Inverted Electrode Geometry for Measuring and Control of Ion Flux-Energy Distribution Functions"

_plasma, doi:10.3390/plasma5030023_

Round 1
Reviewer 1 Report
The manuscript reports a comprehensive and well-designed method by using energy selective mass spectrometry (ESMS) to measure IEDFs in a low pressure plasma chamber with inverted electrode geometry. The study might be attractive for ion surface treatments. Some comments and questions are listed below:
1. The abstract needs more concise.
2. Please provide a reference of ion transmission probability functions between low mass and high mass in mass spectrometry.
2. Figure 3 and 4 show wave shape on IEDFs, but Figure 6 doesn't. Could authors explain the reason?
Author Response
Dear reviewer,
Thank you very much for your helpful comments and queries. All of them are answered in the revised version of the text:
- The abstract has been shortened as required.
- Three references have been provided. The range of the exponents in the mass dependence has been broadened to -0,5 to -1 (it was "close to -1" before). The effect on the relative intensities of He and Ar ions is now given as "between 3 and 10" now.
- The reason is explained on the page 8. We have added following sentence to make it more clear: "(the multiple peak structure at lower energies disappeared and the IEDF is dominated by the bimodal peak structure)"
Reviewer 2 Report
The manuscript “Chamber with inverted electrode geometry for measuring and control of ion energy-flux distribution functions”, by Christian Schulze, He Li, Leonie Mohn, Martin Müller and Jan Benedikt, highlights important results due to CPP and ICP plasma methods, in terms of ion flux-energy distribution functions measurements. The manuscript is well written and can be accepted for publications with minor changes.
1. Abstract- General remark: usually in the abstract, acronyms are not inserted. Herein, it should be briefly presented, your idea of the experiments, and what you’ve accomplished. Please let the acronyms only inside the manuscript, and not in the abstract.
2. Line 88: the authors should insert the long name of the “CCP”. What does it mean “CCP”? It’s stand for “Capacitively Coupled Plasma”? Is yes, please insert the long name near the acronym, when you introduce the acronym for the first time in the manuscript.
3. How damaged were the electrodes, after the experiments? Do you have a high sputtering rate, due to Ar usage?
Author Response
Dear reviewer,
thank you very much for your helpful comments and queries. All of them are answered in the revised version of the text:
- The acronyms have been removed and the abstract has been shortened.
- The long name of the CCP abbreviation has been added in line 88.
- A comment about the sputtering rate "Even though sputtered metal ions are detectable in the ion flux, the sputtering damage to the electrode is negligible as the sputter rate is low for the typical used ion energies in the range of up to about 140 eV in this work. The sputtered material also coats the quartz cylinder, which shows a reflective but still semitransparent metal layer after several hours of operation. To ensure an unhindered rf field transmission, the cylinder was cleaned with nitric acid from time to time." has been added to the manuscript on the page 8.